# Evaluation of Local Retinal Function in Light-Damaged Rats Using Multifocal Electroretinograms and Multifocal Visual Evoked Potentials

**DOI:** 10.3390/ijms242216433

**Published:** 2023-11-17

**Authors:** Yuya Takita, Eriko Sugano, Kazuya Kitabayashi, Kitako Tabata, Akito Saito, Takanori Yokoyama, Reina Onoguchi, Tomokazu Fukuda, Taku Ozaki, Lanlan Bai, Hiroshi Tomita

**Affiliations:** Laboratory of Visual Neuroscience, Graduate Course in Biological Sciences, Iwate University Division of Science and Engineering, 4-3-5 Ueda, Morioka 020-8551, Iwate, Japan; g0322123@iwate-u.ac.jp (Y.T.); sseriko@iwate-u.ac.jp (E.S.); g0322049@iwate-u.ac.jp (K.K.); ktabata@iwate-u.ac.jp (K.T.); s0220015@iwate-u.ac.jp (A.S.); s0220027@iwate-u.ac.jp (T.Y.); g0322033@iwate-u.ac.jp (R.O.); tomof009@iwate-u.ac.jp (T.F.); tozaki@iwate-u.ac.jp (T.O.); bailanlan2010@gmail.com (L.B.)

**Keywords:** multifocal recordings, electroretinograms, visual evoked potential, light-induced photoreceptor degeneration

## Abstract

Electroretinograms (ERGs) are often used to evaluate retinal function. However, assessing local retinal function can be challenging; therefore, photopic and scotopic ERGs are used to record whole-retinal function. This study evaluated focal retinal function in rats exposed to continuous light using a multifocal ERG (mfERG) system. The rats were exposed to 1000 lux of fluorescent light for 24 h to induce photoreceptor degeneration. After light exposure, the rats were reared under cyclic light conditions (12 h: 5 lux, 12 h: dark). Photopic and multifocal ERGs and single-flash and multifocal visual evoked potentials (mfVEPs) were recorded 7 days after light exposure. Fourteen days following light exposure, paraffin-embedded sections were prepared from the eyes for histological evaluation. The ERG and VEP responses dramatically decreased after 24 h of light exposure, and retinal area-dependent decreases were observed in mfERGs and mfVEPs. Histological assessment revealed severe damage to the superior retina and less damage to the inferior retina. Considering the recorded visual angles of mfERGs and mfVEPs, the degenerated area shown on the histological examinations correlates well with the responses from multifocal recordings.

## 1. Introduction

Retinitis pigmentosa (RP) [1] and age-related macular degeneration (AMD) [2] are classified as progressive diseases that cause severe visual impairment. RP causes night blindness and loss of the visual field in the early stages of the disease, ultimately leading to blindness. Various causative genes have been identified, contributing to the complexity of the mechanisms involved in disease progression, and new types of gene mutations have been discovered [3]. A characteristic feature of AMD is the formation of extracellular drusen between retinal pigment epithelial (RPE) cells and Bruch’s membrane. Oxidative stress and the decline in RPE function, leading to the induction of inflammatory responses, are considered the primary factors in AMD’s pathogenesis [4,5]. Although some affected genes contribute to disease progression [6], gene mutations have also been identified [7]. AMD has two types: exudative AMD, caused by neovascularization (wet AMD), and atrophic AMD, in which the macula itself degenerates (dry AMD) [8,9]. The increasing number of patients afflicted by photoreceptor degenerative conditions like RP and AMD due to ageing underscores the demand for treatments and strategies to oversee the progression of photoreceptor degeneration [10,11].

Moreover, understanding the mechanisms of photoreceptor degeneration in these diseases to develop therapeutic methods is crucial. Various animal models with photoreceptor degeneration, including transgenic animals, have been developed. Photoreceptor degeneration models have used the carcinogen N-methyl-N-nitrosourea (MNU) [12,13,14], continuous light-damaged (LD) models [15,16,17,18], and animals with photoreceptor degeneration caused by genetic mutations, such as Royal College of Surgeons (RCS) rats [19,20,21], transgenic rats [22], mice [23], and rabbits [24]. Among these, a variation in degeneration exists between the superior and inferior retina, with a more pronounced degeneration observed in the superior retina [25,26,27].

An electroretinogram (ERG), used to evaluate visual function in animals, is a measurement technique that enables the evaluation of retinal function [28]. ERG can predominantly evaluate the rod cell function of the entire retina by applying light stimulation to the eye; however, normal ERG cannot identify the location of degeneration in cases of localized retinal degeneration. In contrast, in multifocal ERGs (mfERGs), the local response in the retina divided into hexagons can be recorded by randomly stimulating each area, making it possible to detect localized photoreceptor degeneration [29,30,31]. Visual evoked potentials (VEPs) are used to evaluate the visual system, including the visual cortex of the retina [32]. Like the ERG, these VEPs also record the response of the entire retina; therefore, examining the local retinal response is challenging. Multifocal VEPs (mfVEPs) can be recorded using the same stimulation method as mfERGs, providing information on the visual pathway from each retinal area.

Many reports have shown electrophysiological changes after light-induced damage in rat models evaluated using ERG; however, the ERGs provided a whole-retinal response and could not determine the function of the local retinal area. In this study, we recorded mfERGs and mfVEPs in an LD rat model. Different electrophysiological responses depending on the retinal areas were detected in mfERG and mfVEP, which correlated well with histological changes. We demonstrated that mfERG and mfVEP are useful for evaluating local retinal function.

## 2. Results

### 2.1. Recordings of ERGs and mfERGs on the LD Rats

ERGs and mfERGs were measured one week after LD. Before LD, typical waveforms of ERGs, a- and b-waves, including oscillatory potential (OP), were recorded even at a stimulus intensity of 0.01 cd·s/m^2^ (Figure 1A). The amplitudes of the a- (Figure 1B) and b-waves (Figure 1C) after LD were significantly smaller than those before LD. We observed the peak response around the centre of the retina before LD, and the responses became weak towards the peripheral areas (Figure 1D). Comparison of the three-dimensional (3D) density plots before and after LD showed an overall change from warm to colder colours, especially in the superior retina (Figure 1E). The waveforms obtained from multiple discrete areas were depicted, with the amplitudes of the waveforms in all, superior, and inferior regions grouped and quantified. In all regions, the amplitudes after degeneration were significantly smaller than those before degeneration. When the amplitude of the superior retina averaged by 15 hexagons surrounded with red (Figure 1D) was compared to that of the inferior retina surrounded with gold, the amplitude of the superior retina was significantly larger than that of the inferior retina before LD (Figure 1E). After LD, no significant difference was observed in the amplitude between the superior and inferior retinas, and the remaining response from the superior retina tended to be smaller than that from the inferior retina (Figure 1F).

### 2.2. Recordings of VEPs and mfVEPs on the LD Rats

VEP and mfVEP were measured one week after LD concurrently with the ERG. Typical waveforms of VEPs were recorded from rats before LD, and a dramatically reduced response was observed in LD rats (Figure 2A). The amplitudes in the LD rats were significantly smaller than those before LD (Figure 2B). Distinct responses of mfVEPs within the whole retina were mostly observed in the central areas, and 3D density plots (Figure 2C) indicated that the amplitudes from the central area and superior parts were reduced after LD. The retinal areas were divided into three groups: the whole retina, superior regions, and inferior regions, and the amplitudes of each group were calculated. The grouped amplitudes of the whole and superior regions were significantly smaller than those before LD (Figure 2D). No significant differences were observed in the grouped amplitudes of the inferior regions. The remaining responses of grouped amplitudes in the superior regions were greater than those in the inferior regions (Figure 2E).

### 2.3. Histological Evaluation of the LD Model

Paraffin-embedded sections prepared from rat eyes two weeks after LD were stained with haematoxylin and eosin (HE) to assess retinal thickness. Compared to the controls, the LD group showed reduced retinal thickness in the inner limiting membrane–retinal pigment epithelium (ILM-RPE), outer nuclear layer (ONL), and photoreceptor layers (Figure 3). Severe photoreceptor degeneration was observed, particularly in the superior retina.

### 2.4. Correlation of Electrophysiological and Histological Evaluations

To evaluate the correlation between electrophysiological and histological evaluations, we compared the percentage of recorded amplitudes in the mfERG and mfVEP after LD to those before LD as a marker of residual photoreceptors, with retinal thickness obtained from HE staining. The viewing angle was set at 40 degrees for the recordings of mfERG and mfVEP, which corresponded to the retinal area within approximately 2.3 mm of the optic nerve head. We plotted the percentage of residual amplitudes from the inferior (group 1) to the superior regions (group 7) (Figure 4A) on a graph based on the thickness of the ONL at positions of −2.5, −1.725, −1.15, −0.575–0.575, 1.15, 1.725, and 2.5 mm, respectively. A noticeable trend showed that the superior retina experienced significant impairment in ONL and ONL-RPE. Comparing the remaining response obtained from mfVEP and retinal thickness obtained from HE staining, the superior retina tended to be particularly impaired in the ONL and ONL-RPE, similar to the results for mfERG (Figure 4B).

## 3. Discussion

Photoreceptor degeneration through LD is a well-established model that is used in research to screen treatments and explore the mechanisms of photoreceptor degeneration. ERGs and VEPs are commonly used to investigate the physiological functions of the whole retina. Differences in degeneration rates between the superior and inferior parts of the retina are caused by light-induced photoreceptor degeneration [25,26,27]. The data obtained from the mfERGs and mfVEPs showed a dramatically decreased response in the superior retina compared to that in the inferior retina. Fourteen days after light exposure, we evaluated the loss of photoreceptor cells on the histological evaluations. The thickness of the photoreceptor cells seemed to correlate well with the responses of mfERGs and mfVEPs recorded 7 days after light exposure, indicating that the recording of mfERG and mfVEP may be a good marker for evaluating retinal area photoreceptor degeneration.

The stimulus frequencies in the mfERG and mfVEP depend on the number of hexagonal split patterns. We used 37 split patterns with a stimulus frequency of 75 Hz. Rod cells seem to have difficulty responding to a high frequency of stimulation because the rat retina does not have a fovea, as in humans. However, the 3D response density plots of the mfERG from the retina before LD (Figure 1D) showed the highest response at the centre of the retina. Previous reports have indicated that the distribution of rods and cones in the rodent retina is not uniform, with cones being particularly abundant in the central region of the retina, even though their structural characteristics differ significantly from those in humans [33,34]. Moreover, mfERG in rodents includes mixed responses of rods and cones. We could not detect the optic nerve head in the data of mfERGs. The optic nerve head may have been present at the centre of the recording area. However, the defective area of response originating from the optic nerve head did not exist around the centre of the recording area because the recording area of the central hexagon is approximately 1.15 mm in width, much larger than the 300 μm optic nerve head diameter [33].

The 3D response density plots show a difference between the superior and inferior retinas in mfERG pre- and post-photoreceptor degeneration. In particular, the superior retina showed a reduced response (Figure 1D,E). These results indicated that functional changes in the superior retina occurred in the early phase of light-induced photoreceptor degenerations, as shown as a characteristic of the LD model in the histological examinations. The degree of damage is reported to be more severe in the superior retina in the LD; however, the complete mechanism is unclear. One of the reasons for severe degeneration of the superior retina is low expression of basic fibroblast growth factor (bFGF), a neuroprotective factor [34,35]. bFGF, whether administered endogenously [36,37,38] or exogenously through intravitreal injections or other means, has a protective effect on the retina [39,40,41]. Furthermore, bFGF was induced in the retina after LD [35,42,43], and the level of expression differs between the superior and inferior retinas, with more expression and protection in the inferior retina and less expression in the superior retina [34,35]. Different retinal responses between the superior and inferior retinas were recorded using mfERGs with well-defined region-dependent light damage. Some experiments on the effects of drugs using LD models have reported that drugs were observed to be effective on the inferior retina and not on the superior retina, causing severe damage [17]. Detecting local degeneration in the early phase of LD using ERG is challenging. Moreover, mfERGs can detect specific features of the LD model, allowing us to assess each area of the retina electrophysiologically. Besides conventional histological evaluation, simultaneous electrophysiological evaluation, as in mfERG in this study, allows a more detailed evaluation of each area of the retina from multiple viewpoints and enables consideration of the effects of drugs from a new perspective.

The ERG (b-wave, 10.0 cd·s/mm^2^) and VEP were attenuated to 1.07 ± 0.45% and 32.02 ± 6.79% after the LD, respectively. The remaining responses of VEPs were much lower than those of ERGs, indicating that the responses of the upper-level neurones emerged from the integrated response of the lower-level neurones [44]. When the responses of mfERG were divided into the superior and the inferior parts of the retina, the remaining response were 33.16 ± 3.95% and 44.89 ± 9.45%, respectively. Those of mfVEP were 47.71 ± 12.71% and 114.94 ± 21.14%, respectively, indicating that the inferior retina was less affected by the LD.

We histologically evaluated the degree of degeneration in the retina. Compared to the control group, the LD group showed thinning of the ILM-RPE, ONL, and ONL-RPE, especially in the superior retina. This is consistent with the characteristics of the LD model reported in other studies [17,25,26]. The measurement area of the mfERG and mfVEP, set at the viewing angle of 40 degrees in this study, was the region with a diameter of approximately 2.3 mm from the optic nerve head. The remaining response was calculated from the mfERG and mfVEP results, which showed a similar trend to ONL thickness in the histological examinations (Figure 4).

In this study, we used mfERG and mfVEP to identify the areas degenerated by LD. Another use of multifocal recordings is in gene therapy. Recently, channelrhodopsin (ChR) has been used in gene therapies, and various ChR genes have been developed [45,46,47,48]. The goal of ChR gene therapy is to restore vision and to protect the retinal neurones by expressing the ChR in the retinal ganglion cells [20,49], on-bipolar cells [50,51], and photoreceptor cells [52]. Restoring vision and protecting retinal cells would be the results at sites where the ChR gene is transduced. We could determine the functionally recovered area using mfVEP. Additionally, evaluating the correlation between mfVEP amplitudes and gene expression areas, in conjunction with fluorescent fundus photography to assess gene expressions [53,54,55], could make multifocal recordings a valuable tool for investigating retinal local function.

## 4. Materials and Methods

### 4.1. Animals

Eight-week-old male Wistar rats (CLEA Japan Inc., Tokyo, Japan) were used in this study. All rats were housed under conditions of cyclic light (on 8:00 a.m., 5 lux; off 8:00 p.m., 0 lux) at 23 ± 1 °C. Rats were fed laboratory chow ad libitum with free access to water. All the animal experiments were performed in accordance with the ARVO Statement for the Use of Animals in Ophthalmic and Vision Research and the Guidelines of the Iwate University Animal Experimentation Committee on Animals in Research.

### 4.2. Induction of Light Damage

Photoreceptor degeneration was induced as previously described [17]. Briefly, rats were dark-adapted for 24 h before light exposure. Following dilation of the pupils with tropicamide (Midrin-P, Santen Pharmaceutical Co., Ltd., Osaka, Japan), the rats were placed in a lightbox (041001, NK system, Osaka, Japan) at 1000 lux of light for 24 h.

### 4.3. Recording of ERGs and VEPs

Seven days after LD, ERGs and VEPs were recorded using PuREC (Mayo Co., Aichi, Japan) using previously reported methods [17]. Briefly, the rats were dark-adapted overnight and anaesthetized via intramuscular injection of a mixture of ketamine hydrochloride (45 mg/kg; Daiichi Sankyo Propharma Co., Ltd., Tokyo, Japan) and xylazine (4.5 mg/kg; Elanco Japan, Tokyo, Japan). Local anaesthesia was induced by a topical application of an eye drop (Benoxil ophthalmic solution 0.4%; Santen Pharmaceutical Co., Ltd.), and the pupil was dilated with a 10-fold dilution of tropicamide (Santen Pharmaceutical Co., Ltd.), and then the eyes were covered with small contact lenses with a gold wire loop filled with a small amount of hydroxyethyl cellulose eye solution (Scopisol 15^®^; Senju Pharmaceutical Co., Ltd., Osaka, Japan). The reference and ground electrodes were placed on the tongue and feet, respectively. Stimulus intensities (0.01, 3.00, and 10.0 cd·s/m^2^) were set according to the standard method recommended by the International Society of Clinical Electrophysiology of Vision (ISCEV) [28]. High- and low-path filters were set at 0.3 Hz and 500 Hz, respectively. After measurements, the eyes of the rats were washed with physiological saline.

Electrodes were implanted in the visual cortex of rats 7 days before recording the VEP. The stimulus intensity was set at 3.00 cd·s/m^2^, as recommended by the ISCEV standard method [32]. The high- and low-path filters were set to 1 and 500 Hz, respectively. The VEP responses were consecutively measured 200 times, and the waveforms were averaged.

### 4.4. Recording of mfERGs and mfVEPs

We used mRec (ACrux Inc., Iwate, Japan) to record the mfERGs and mfVEPs with the same conditions as the ERGs and VEP recordings, except for the stimulus pattern. The rat, whose body and neck angle were adjusted such that the centre of the rat’s eye was on the extension of the centre portion indicated by a red cross on the screen, was placed at the measuring stage. The distance from the screen to the rat’s eye was set at 19 cm, resulting in a 40-degree viewing angle. Thirty-seven hexagonal split patterns were displayed randomly for retinal stimulation at 75 Hz. The amplitude of individual hexagons was calculated using mREC software ver. 1.02. In detail, the first negative wave and the first positive wave following the first negative wave were selected as N1 and P1, respectively. The amplitude of N1-P1 was divided by the area of the hexagon, represented as the individual amplitude of the hexagon.

### 4.5. Histological Studies of the Retinas with HE Staining

The rats were sacrificed with an overdose of carbon dioxide (CO_2_) 2 weeks after LD. The eyes of the rats were enucleated, fixed, and embedded in paraffin. Five-micrometre sections were cut along the vertical meridian and stained with HE to compare all areas of the retina in the superior and inferior hemispheres. The thickness of retinal layers was measured at 500 μm intervals in the vertical direction, including the optic nerve head.

### 4.6. Statistical Analysis

All statistical analyses were performed using GraphPad Prism (MDF, Tokyo, Japan). The analyses involved paired *t*-tests, unpaired *t*-tests, and Dunn’s multiple comparison test.

## Figures and Tables

**Figure 1 ijms-24-16433-f001:**
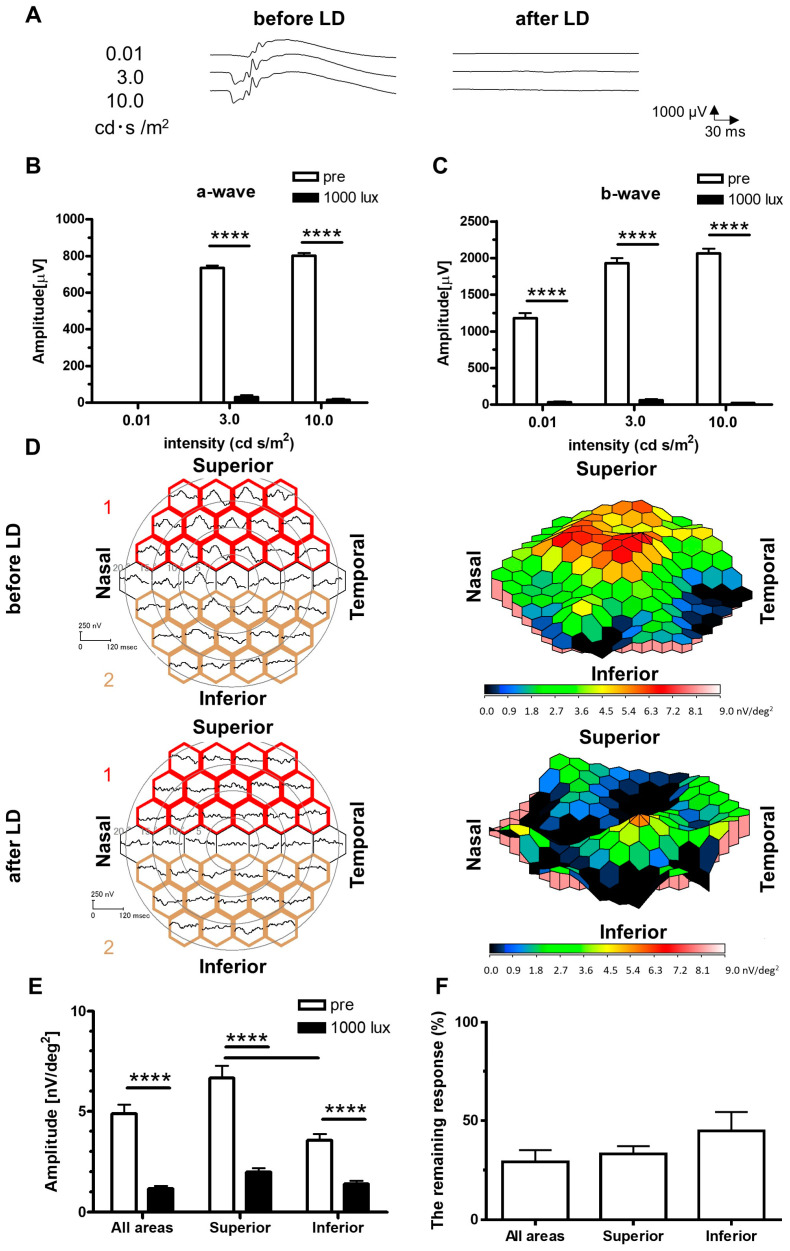
Recordings of ERGs and mfERGs on the LD rats. Typical waveforms of ERGs before and after LD (**A**). Amplitudes of the a- (**B**) and b-waves (**C**) were significantly reduced after LD. Data are presented as the mean ± SEM (pre: *n* = 8, 1000 lux: *n* = 8, paired *t*-test, **** *p* < 0.0001). Averaged waveforms and 3D density plots, a visualization of the amount of retinal response per unit area, are shown in the left and right panels, respectively (**D**). The amplitudes recorded from the mfERG were divided into three groups: the whole retina, the superior retina surrounded with red (group 1), and the inferior retina surrounded with gold (group 2). Comparisons of the amplitudes in the superior and inferior retinas are shown in (**E**). Data are presented as the mean ± SEM (pre: *n* = 16, 1000 lux: *n* = 16, paired *t*-test, **** *p* < 0.0001). The remaining response, calculated by dividing the amplitude value after LD in the mfERG by the pre-value, is shown in (**F**). Data are presented as the mean ± SEM (all areas, superior, inferior: *n* = 16, Dunn’s multiple comparison test). mfERGs, multifocal electroretinograms; LD, light-damaged; SEM, standard error of the mean; 3D, three-dimensional.

**Figure 2 ijms-24-16433-f002:**
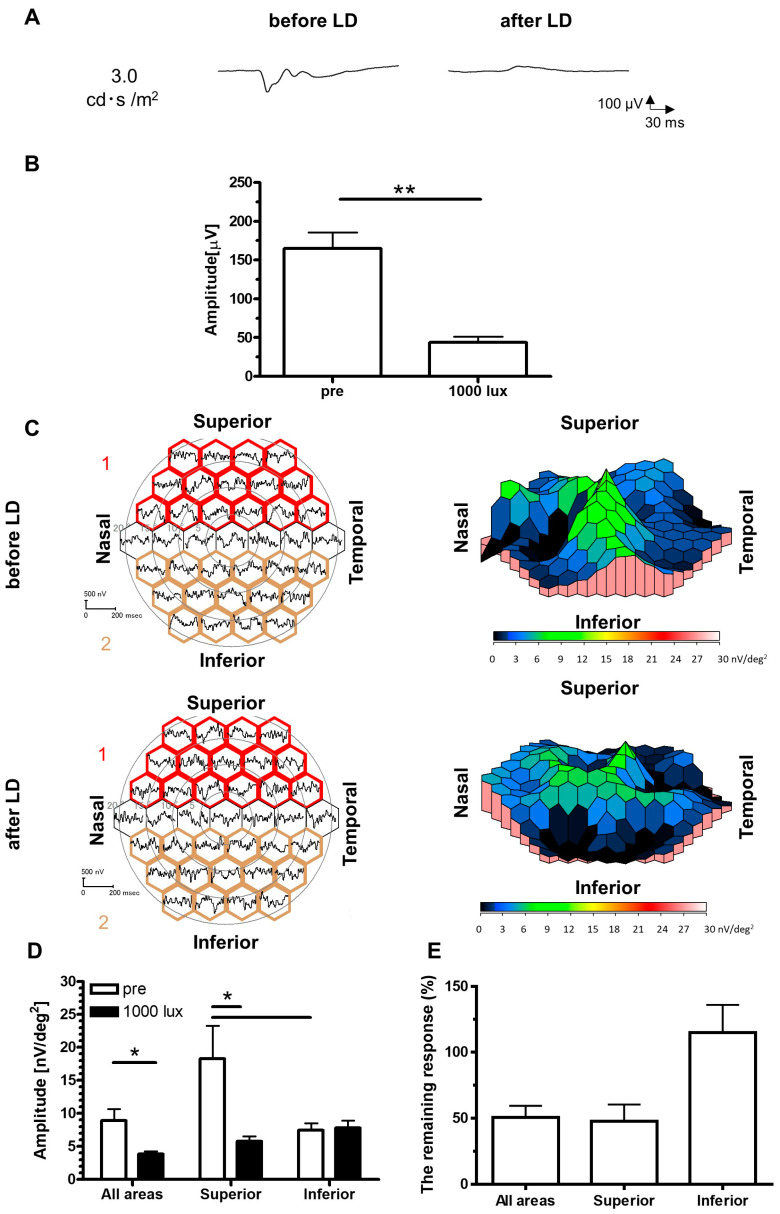
Recordings of VEPs and mfVEPs on the LD rats. Typical waveforms of VEPs before and after LD (**A**). Comparison of the recorded amplitudes before and after LD (**B**). Data are presented as the mean ± SEM (pre: *n* = 8, 1000 lux: *n* = 8, paired *t*-test, ** *p* < 0.01). Averaged waveforms and 3D density plots, a visualization of the amount of retinal response per unit area, are shown in the left and right panels, respectively (**C**). The amplitudes of the mfVEP were divided into three groups: the whole retina, the superior retina surrounded with red (group 1), and the inferior retina surrounded with gold (group 2). Comparisons of the amplitudes in the superior and inferior retinas are shown in (**D**). Data are presented as the mean ± SEM (pre: *n* = 7, 1000 lux: *n* = 7, unpaired *t*-test, * *p* < 0.05). The remaining response was calculated by dividing the amplitude value after LD in mfVEP by the pre-value, as shown in (**E**). Data are presented as the mean ± SEM (all areas, superior, inferior: *n* = 7, Dunn’s multiple comparison test). mfVEP, multifocal visual evoked potential; LD, light-damaged; SEM, standard error of the mean; 3D, three-dimensional.

**Figure 3 ijms-24-16433-f003:**
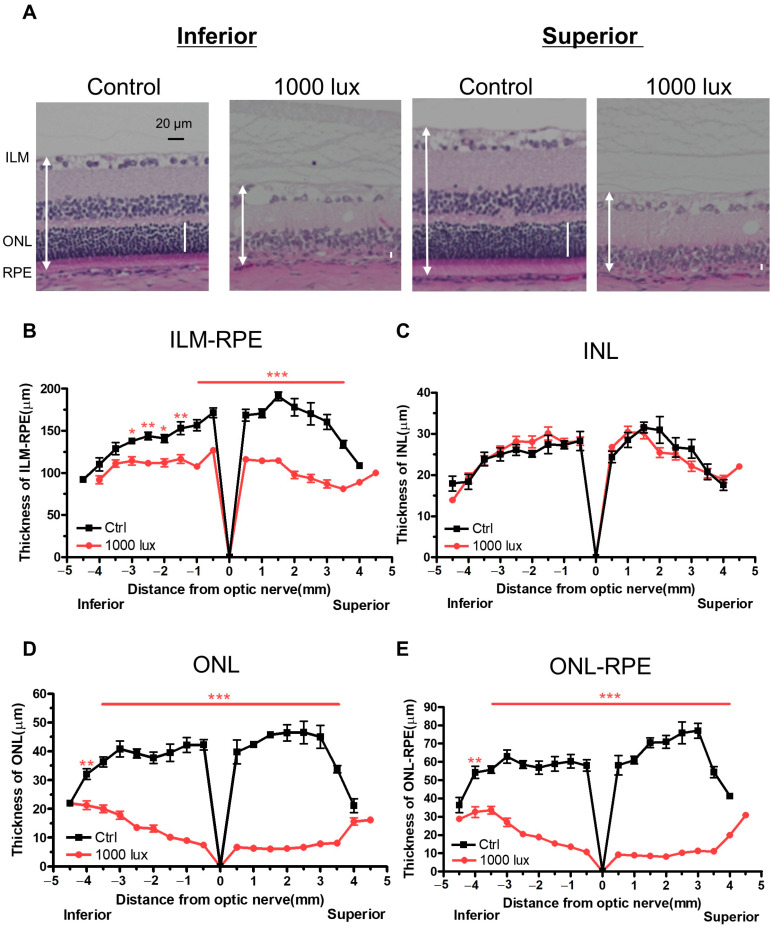
Histological evaluation of retina two weeks after LD. Typical HE images of the superior and inferior retinas are shown in (**A**). The white two-direction arrow and the white vertical bar indicate the region to ILM from RPE and ONL, respectively. Thickness of the ILM-RPE (**B**), INL (**C**), ONL (**D**), and ONL-RPE (**E**) in rats with and without 1000 lux LD. Data are presented as the mean ± SEM values (control: *n* = 4, 1000 lux: *n* = 24, unpaired *t*-test, * *p* < 0.05, ** *p* < 0.01, *** *p* < 0.0001). HE, haematoxylin and eosin; ILM, inner limiting membrane; RPE, retinal pigment epithelium; ONL, outer nuclear layer; SEM, standard error of the mean; LD, light-damaged.

**Figure 4 ijms-24-16433-f004:**
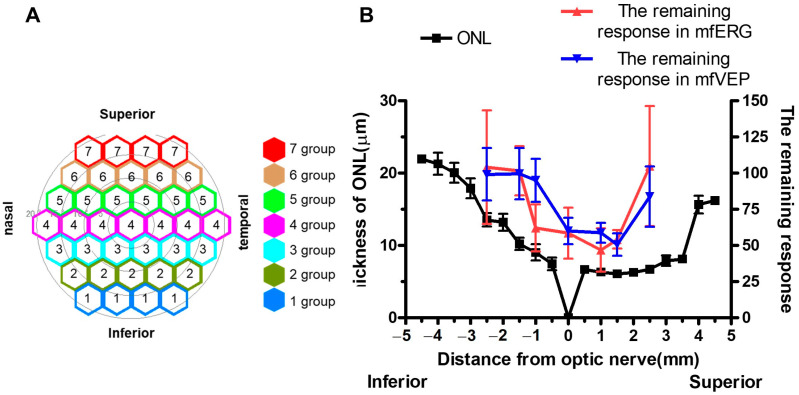
Correlation of histological evaluation with mfERG and mfVEP. As shown in (**A**), the averaged amplitude of each group from the inferior to the superior regions was calculated, and the values as a percentage of the pre-recorded amplitude were overlaid on the histological thickness (**B**). Data are presented as the mean ± SEM (1 group~7 group of mfERG: *n* = 16, 1 group~7 group of mfVEP: *n* = 7, ONL: *n* = 24). The position of the retina in each group of mfERG and mfVEP is 1 group: −2.5 mm, 2 group: −1.5 mm, 3 group: −1.0 mm, 4 group: 0 mm, 5 group: 1.0 mm, 6 group: 1.5 mm, 7 group: 2.5 mm. SEM, standard error of the mean; mfERG, multifocal electroretinogram; mfVEP, multifocal visual evoked potential; ONL, outer nuclear layer.

## Data Availability

The datasets used and/or analysed in the current study are available from the corresponding author upon reasonable request.

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
