# Peer review of "Evaluation of Local Retinal Function in Light-Damaged Rats Using Multifocal Electroretinograms and Multifocal Visual Evoked Potentials"

_ijms, 2023, doi:10.3390/ijms242216433_

Round 1

Reviewer 1 Report

Comments and Suggestions for Authors

In this manuscript light-induced retinal damage is studied using multifocal electroretinogram (mfERG) and multifocal visual evoked potential (mfVEP) followed by histopathological analysis of the damaged retinas. Overall, it is a nice proof-of-principle study, however, I have some concerns about the data presented.

First the Authors determine the reduction in ERG waveform amplitudes following light damage, calculating from multifocal (mf) ERG data for the whole retina, the superior retina, and inferior retina, and refer to it as “reduction rate”. The use of “rate” (ln 91, 116, Fig. 1F) is inappropriate, as the “rate” normally refers to a change over time. When a parameter (i.e. mfERG amplitude) measured and compared only at two timepoints, change in amplitude/magnitude should be used.

Related point 1: whereas the way of comparison of a- and b-wave amplitudes before and after light damage (Fig 1A, 1B. 1C) is clear, Authors do not describe at all how they measured the mfERG amplitudes (beyond saying that waveforms were grouped and quantified, ln 85).

Related point 2: Authors state: “When the amplitude of the superior retina averaged by 15 hexagons surrounded with red (Figure 1D) was compared to that of the inferior retina surrounded with gold, the amplitude of the superior retina was significantly larger than that of the inferior retina before LD (Figure 1E).” (ln 86-89). However, no statistical comparison is show in Figure 1E between the superior and inferior retina before LD. Similarly, Authors state: “After LD, no significant difference was observed in the amplitude between the superior and inferior retinas, and the reduction rate of the amplitude from the superior retina was greater than that from the inferior retina (Figure 1F).”  (ln 89-92) Again, there is no statistical comparison shown for the amplitude of the superior and inferior retinas after LD. As for the second half of the sentence (i.e. “…, and the reduction rate...”), see below.

Figure 1E does not seem to line up with Fig 1F. For example: the “pre” amplitude of “All areas” is roughly 5x of that after the light damage (LD) (Fig 1E), but only ~25% “reduction rate” is shown in Fig. 1F.  The normalized “rate of reduction” for the whole and superior retina in Fig 1F is the smaller than that of the inferior retina, although the cumulative “raw data” (i.e. not normalized) in Fig 1E appears to be the other way around. The first chapter of the results as well as the alignment of text with the figures need substantial revision.

In the next chapter of the result using “rate” for VEPs and mfVEPs changes is also inappropriate, as above. The method of mfVEP quantification is not communicated. Fig 2D does not line up with 2E: For example Fig 2D show the reduction of mfVEP amplitude (however it was measured/calculated from ~11 nV/deg^2 before LD to about 4 nV/deg^2 after LD, which is a bit more than 60% reduction in average. Yet, the corresponding bar of Fig 2E show 50% average “reduction rate of amplitude”. Similarly, the average mfVEP amplitude in the superior retina from ~17 nV/deg^2 went to 6 nV/deg^2 after LD, wich is again an approximatel 65% reduction, but 50% reduction is shown in Fig 2E. ­­There is no LD-mediated reduction in mfVEP amplitude (Fig 2D) yet Fig 2E show 100% average “reduction rate of amplitude”. Whether or not the presentation of the normalized data in Figure 2E is correct, it is hard to believe that Dunn’s multiple comparison test could not detect significant difference in the data plotted in Fig 2E (i.e. 50% in average vs. 100%, with about 10% error).

Along the same lines, in the Discussion the Authors are taking about 114.94 ± 21.14% reduction of the mfVEP (ln 220). I think something is fundamentally wrong with the quantification, but because it has not been explained at all, it is difficult to point it out.

Author Response

Thank you for your detailed review and for giving us suggestions.

A kind thorough explanation is beneficial for us to revise our manuscript.

I revised our manuscript following the reviewer’s comments.

I would appreciate it if you re-review our manuscript.

Reviewer 2 Report

Comments and Suggestions for Authors

The manuscript ‘Evaluation of Local Retinal Function in Light-Damaged Rats Using Multifocal Electroretinograms and Multifocal Visual Evoked Potentials’ provides interesting and novelty information about the use of tERGs and mfERG as well as mfVEPs in the light-damage model. Despite there being some controversies with this animal model in the ophthalmology area, the manuscript is well-written and the results/discussion are in accordance with the objectives planned and well-justified by the authors. Therefore, I consider that the manuscript is prepared to be published. 

Please see attached for additional comments.

Author Response

Thank you for your detailed review and for giving us positive responses. A kind thorough explanation is beneficial for us to revise our manuscript.

I revised our manuscript following reviewer 1's comments.

I believe that the revised manuscript become more clear because some of reviewer 1's comments include your question.

I would appreciate it if you re-review our manuscript.

Round 2

Reviewer 1 Report

Comments and Suggestions for Authors

My concerns raised against the original version of the manuscript were addressed during the revision.